# Measures of Adrenal and Gonadal Hormones in Relation to Biological and Management Factors among Captive Red Pandas in Indian Zoos

**DOI:** 10.3390/ani13081298

**Published:** 2023-04-10

**Authors:** Aamer Sohel Khan, Janine L. Brown, Vinod Kumar, Govindhaswamy Umapathy, Nagarajan Baskaran

**Affiliations:** 1Mammalian Biology Lab, Department of Zoology and Wildlife Biology, Anbanathpuram Vahaira Charity (A.V.C.) College (Autonomous), Mannampandal, Mayiladuthurai 609305, Tamil Nadu, India; khannaamirsohel@gmail.com; 2Smithsonian National Zoo and Conservation Biology Institute, Center for Species Survival, Front Royal, VA 22630, USA; brownjan@si.edu; 3Laboratory for the Conservation of Endangered Species, CSIR-Center for Cellular and Molecular Biology, Habsiguda, Hyderabad 500007, Telangana, India; vinod@ccmb.res.in (V.K.); guma@ccmb.res.in (G.U.)

**Keywords:** captive breeding, endangered, red panda, reproductive hormone, stress hormone, welfare

## Abstract

**Simple Summary:**

Red pandas are a threatened species, and zoos worldwide are working to conserve the species through international captive breeding programs. However, information on the physiology of captive red pandas is limited, which is hampering efforts to increase reproductive success and meet captive breeding goals. This study measured concentrations of fecal glucocorticoid (fGCM), progestagen (fPM), and androgen (fAM) metabolites in relation to environmental and biological factors among captive red pandas housed at three Indian zoos. Data revealed that fGCM concentrations were influenced by social time together (−), visitor numbers (+), number of nests (+), frequency of feedings (−), log density (−), and enclosure area (+). The fPM results suggested that enclosure area negatively affected concentrations in females. By contrast, fAM in males was not affected by these factors. This study suggests that employing more frequent feeding schedules and controlling the number of visitors may have positive effects on the welfare of red pandas, while higher fGCM and lower fPM in larger enclosures might be related to the limited enrichment and hiding spaces in those areas.

**Abstract:**

Animals in human care are affected by stressors that can ultimately reduce fitness. When reproduction is affected, endangered species conservation programs can be severely compromised. Thus, understanding factors related to stress and reproduction, and related hormones, is important to ensure captive breeding success. Red pandas (Ailurus fulgens) are endangered, and populations in the wild are threatened with extinction. A global captive breeding program has been launched to conserve the species with the goal of reintroduction. However, there is little information on how stressors impact physiological aspects of the species. This study measured fecal glucocorticoid (fGCM), progestagen (fPM), and androgen (fAM) metabolite concentrations in 12 female and 8 male red pandas at 3 zoos in India to determine predictors of adrenal and gonadal steroid activity, and the influence of fGCM on reproduction. Based on the generalized linear mixed model (GLMM), fGCM concentrations were positively correlated with the number of visitors, number of nests and enclosure areas, and negatively related to frequency of feedings, log density, and social time, while fPM concentrations were negatively associated with enclosure areas. A confounder for enclosure areas and number of nests was the fact that these spaces were relatively barren, with limited hiding spaces, compared to the smaller enclosures. By contrast, no significant relationships were found for fAM, perhaps due to the smaller sample size. A negative relationship between fGCM and fPM was observed, indicating increasing adrenal hormones may decrease reproductive function among female red pandas. Results suggest that zoo management should consider increasing feeding frequency, providing larger enclosures with more enrichment and more nests in larger spaces, and regulating visitor numbers to support good welfare and potentially improve reproductive fitness of red pandas in captivity.

## 1. Introduction

The red panda is an endemic species to the eastern Himalayas from Nepal, India, Bhutan, and China extending up to Myanmar, and lives at altitudes of 1500–4000 m [1,2,3]. The species is habitually nocturnal and belongs to a monophyletic family, Ailuridae, which consists of two sub-species, Ailurus fulgens fulgens and Ailurus fulgens styani [1]. Red pandas are currently threatened throughout much of their range [2] due to poaching and destruction of bamboo forests [1,4], and wild populations are on a declining trend with less than 10,000 mature individuals remaining [2]. Worldwide efforts to protect the species include global captive breeding programs with the potential for reintroduction [2,5,6]. Nevertheless, populations are not stable in captivity [3,7], owing to low reproductive success and poor infant survival [7]. The reasons for these problems are unknown, but could be related to stress and compromised welfare [8].

Globally, there are 92 institutions housing more than 300 red pandas [2], with a total of 36 individuals in India. Studbook data suggest that the breeding success of red pandas across institutions varies considerably [9]. Environmental factors and general husbandry practices can exert stress in captive animals [10], although responses are often species-specific [8]. Environmental and biological factors such as presence of visitors, enclosure size, diet, enrichment, age, and sex can affect stress responses in numerous species [10,11,12,13]. In red pandas, it has been shown that high ambient temperatures; small enclosure size; improper enclosure substrates; lack of hiding places, nest boxes, and climbing structures; and exposure to visitors all may contribute to stress and reduced reproductive success in captivity [5,9,14]. Climate, in particular, may exert stress in red pandas because of warmer weather in places where many captive populations are kept [2,5,9]. Finally, offspring survival is higher in younger as compared to older pandas [15], indicating parent age may also influence breeding success. 

A few studies have evaluated glucocorticoids and reproductive hormones in captive red pandas [7,16,17,18], but overall, the stress and reproductive physiology of this species is poorly understood, especially with regard to the effects of management factors. In our previous study of Ailurus fulgens fulgens living in three zoos in India, stereotypic behaviours in the form of pacing, tongue flicking, and position circling were observed, which is an indication of poor welfare [19]. That study also found that factors such as log density, number of nests, tree density, and tree height all influenced rates of stereotypy [19]. Furthermore, as stress is known to influence reproduction negatively in other species [8], this study evaluated how environmental and biological factors were related to adrenal and gonadal hormones in the same population of red pandas, assessed in our earlier study [19]. Our hypotheses were that: (i) variations in environmental and biological factors influence fecal glucocorticoid (fGCM), progestagen (fPM), and androgen (fAM) concentrations; and (ii) fGCM concentrations are negatively related to fAM and fPM concentrations. Because captivity is known to influence the fitness of individuals, understanding factors related to stress and reproduction, and measures of related hormones, is important to ensure captive breeding success. The overall goal was to provide empirical data that could be used to improve management strategies, effectively addressing the welfare needs of red pandas in captivity.

## 2. Materials and Methods

### 2.1. Study Sites

This study was carried out in three zoological facilities in India: (1) Padmaja Naidu Himalayan Zoological Park (PNHZP), Darjeeling; (2) Sikkim Himalayan Zoological Park (SHZP), Gangtok; and (3) G. B. Pant High Altitude Zoo (GBPHAZ), Nainital. All zoos participate in the Red Panda Global Species Management Plan. The first captive breeding center in the country for the species was established at PNHZP in 1957. PNHZP had multiple enclosures with varying sizes ranging between 192 m^2^–3068 m^2^ (see Appendix A). The enclosures mimic a wet temperate forest climate with temperature variation from sub-zero in winter to 25 °C during the summer. Animals in this study were maintained in semi-natural open enclosures with large trees and nests. Bamboo was fed in the evening with supplementary diet items (honey, milk, egg, and fruits) in the morning. SHZP housed three individuals in one large open enclosure (2463 m^2^) with only one tree and a nest; the rest of the enclosure was barren. The diet consisted of bamboo in the evening and supplementary foods (honey, milk, egg, fruits, and breads) in the morning. The temperature varied between 4–22 °C. GBPHAZ kept individuals in small, netted enclosures (224 m^2^) with a nest box, but no trees. Bamboo and supplementary diet items (honey, milk, egg, and fruits) were fed together in the morning. The temperature at GBPHAZ fluctuated from 3 to 25 °C. All three zoos were at a similar altitude. Breeding records for each female were obtained by the registrar at each zoo. Pandas were managed either as solitary or in male-female pairs during the breeding season (January–March), termed ‘Sociality’. Females with young cubs were also housed together (see Appendix A). The four biological and nine environmental covariates assessed in the study are defined in Table 1.

### 2.2. Fecal Sample Collection

A total of 40 fecal samples were collected from 20 red pandas (12 females and 8 males) (Appendix A), with 2 samples from each individual over a 2-week interval between December 2017 and April 2018. Entire fresh fecal samples (excluding the outer shiny layer) were collected in the morning [18]. Individual red pandas defecate in the same place in the enclosure, which aided individual identification; experienced zookeepers also helped in individual fecal identification. Immediately after collection, the samples were dried at 70 °C for 24 h in a hot air oven [20]. The dried samples were packed in sterile zip-lock bags and stored at room temperature in a moisture-controlled room. Samples were transferred within 2 weeks to the Laboratory for the Conservation of Endangered Species (LaCONES), CCMB Hyderabad, India, for processing and hormonal analyses. 

### 2.3. Extraction of Fecal Steroid Metabolites

Fecal steroid metabolites were extracted using protocols described earlier [18,21,22]. Approximately 0.2 g of dried fecal powder was boiled in 5 mL of 90% ethanol for 20 min. The samples were centrifuged at 500× *g* for 20 min and the supernatant transferred to a fresh tube. The pellets were re-suspended in 5 mL of 90% ethanol, vortexed for 1 min, re-centrifuged, and both supernatants combined. The supernatants were dried at 40 °C in an oven, re-suspended in 1 mL of absolute methanol, and vortexed for 1 min. Fecal extracts were kept at −20 °C until analyzed by enzyme immunoassay (EIA).

### 2.4. Enzyme Immunoassays

fGCM were quantified using a polyclonal cortisol antibody (R4866, Coralie Munro, University of California, Davis), which had been validated for use in red panda feces [18,23]. The sensitivity of the assay was 1.95 pg/well. The inter- and intra-assay coefficients of variation (CV) were 7.9% and 5.5%, respectively. 

fPM concentrations were quantified using a 5α-pregnan-3α-ol-20-one EIA (polyclonal antibody) validated for red panda feces [18,22,24]. The sensitivity of the assay was 6 pg/well. The inter- and intra-assay coefficients of variation were 9.4% and 7.3%, respectively. 

fAM concentrations were quantified using a validated polyclonal testosterone antibody (R156/7; Coralie Munro, University of California, Davis) [18,23]. The sensitivity of the assay was 1.17 pg/well. The inter- and intra-assay coefficients of variation (CV) were 8.2% and 6.1%, respectively. All EIAs were performed as described previously [25,26,27].

### 2.5. Statistical Analyses

All analyses were conducted using R (Version 13.4.1). Data on dependent factors were first tested for normality; however, fGCM and fPM data were not normal (Shapiro-Wilk test, *p* < 0.05) and could not be normalized by any of the four transformations. fAM data were found to be normal (Shapiro-Wilk test, *p* > 0.05). One fGCM data point was an outlier (198.91 ng/g) and removed from further analysis. We did not include cubs (*n* = 3) in reproductive hormone analyses. To determine potential predictors of fGCM, a generalized linear mixed model (GLMM) in package *lme4* in R was used, with all 13 covariates listed in Table 1. Sociality (either paired or solitary) was not included as the sample sizes between categories were too small for comparison. The GLMM was fitted with all the covariates as fixed effects, with subject and enclosure IDs (Enclosure1, Enclosure2, Enclosure3, etc.) as random effects. We nested sex within subject (Subject/Sex) to account for any sex-based differences in fGCM concentrations among individuals. We also nested subject within enclosure ID (Enclosure ID/Subject) to accommodate for effects of any unknown enclosure variables. The model was treated with restricted (residual) estimated maximum likelihood (REML). A Farrar–Glauber collinearity test was run with packages *mctest* and *olsrr* in R. We found tree density and log density to be co-related with a variance inflation factor (VIF) > 10, and thus one (i.e., tree density) was removed from the GLMM. 

We included fGCM concentration in the GLMM for fPM as a predictor variable. Here we tested predictors in two categories: Enclosure effects–variables associated with enclosure characteristics (mean tree height used, number of nests, enclosure areas, log and tree density) and non-enclosure effects–variables not related to enclosure characteristics (age, social time, ambient temperature, number of visitors, frequency of feed, and bamboo). In the enclosure effects model, subject and enc. ID were included as random effects; in the non-enclosure effects model, only subject was kept as a random effect. Collinearity was tested before performing the GLMM; no collinearity was found for both sets of variables.

Further, to ensure correct interpretation of the relationship between fGCM and fPM, we accounted for confounding variables using analysis of covariance (ANCOVA). Ambient temperature, number of visitors, mean tree height used, tree density, log density, and number of nests were subjected to regression and homogeneity of variance tests (Levene’s test in package *car*). Only tree density was found to be a confounding variable, and so was controlled for in the ANCOVA. 

## 3. Results

Mean fGCM concentration was 40.00 ± 5.16 ng/g (range: 2.84–95.63 ng/g; *n* = 40) with no difference between males (39.21 ± 5.24 ng/g, *n* = 8) and females (40.56 ± 4.49 ng/g, *n* = 12). Among the 13 covariates tested against fGCM using the GLMM, fGCM concentrations decreased with increased social time (housed together), frequency of feedings, and log density, but increased with number of visitors, number of nests, and enclosure areas (Table 2). These six factors explained 74% of the variations in fGCM concentrations in captive red pandas. 

Mean fPM concentration among adult females was 570.67 ± 150.51 ng/g (*n* = 9) and ranged from 15.95–2632.00 ng/g. Among the six variables tested within the enclosure category, enclosure area was negatively related to fPM concentration in the GLMM (Table 3), explaining 47% variation.

The fAM concentrations in adult males averaged 15.87 ± 2.32 ng/g (*n* = 7), ranged from 2.61–34.28 ng/g, and were not related to any tested factors.

Concentrations of fPM were not related to fGCM. However, when controlled for tree density using ANCOVA, the relationship was significant (*β* = −4286, *t* = −4.65, *p* = 1.56 × 10^4^; *F* (3, 20) = 217.6, *ηp^2^* = 0.55, *p* = 1.97 × 10^15^). There was no relationship between fGCM and fAM concentrations in male pandas (*β* = −0.02, *R*^2^ = 0.005, *p* = 0.768).

## 4. Discussion

This study found a number of factors affected adrenal hormone concentrations in zoo-housed red pandas in India. Social time together, visitor number, number of nests, log density, enclosure areas, and frequency of feedings were significant predictors explaining 74% of the variations in fGCM concentrations in males and females. Overall results suggest employing more frequent feeding schedules, providing larger enclosures with adequate enrichment, giving access to a greater number of nests with enough space, and controlling the number of visitors all may have implications for the welfare of red pandas, and these variables must be explored in future research. The higher fGCM in larger enclosures could be due to the lack of enrichment in those spaces, leaving animals exposed with few hiding places. Similarly, the higher concentration of fGCM associated with a greater number of nests could be due to the limited space within the enclosure, which restricted animal movement. These findings emphasize the importance of providing a balanced combination of space and enrichment for animals to effectively cope with stressors and to help ensure the physical and psychological well-being of this species in captivity.

Social time together was found to be negatively associated with fGCM in this study, meaning pandas kept together and for longer periods of time exhibited lower adrenal activity compared to pandas kept together, but for shorter durations. In our previous study, we showed that psychological stress or stereotypy in red pandas was negatively associated with sociality [19], suggesting socializing red pandas with conspecifics in captivity may reduce stress and abnormal behavior. Similarly, physiological results suggest that socializing red pandas for extended periods of time supports good welfare.

Visitor intensity was a significant and positive predictor of fGCM concentrations in this study, indicating an impact of high visitor numbers on physiological function. Visitors are known to influence animal welfare and are one of the potential predictors for animal welfare in confinement [28]. The red panda is nocturnal by nature, so being continuously exposed to large numbers of visitors during the day, up to 162 per hour, might be having an effect on adrenal steroid production. Similar relationships between visitors and fGCM concentrations have been reported in a number of other captive species [29], including Mexican wolves [30], Indian blackbucks [31], Royal Bengal tigers, and Indian leopards [12]. 

The frequency of feeding was one of the key predictors of decreasing trends in fGCM concentrations, indicating the importance of an adequate, but also variable, food supply on the welfare of captive red pandas. A nutritious and balanced diet is essential for the survival of all organisms, and for animals in captivity, unlike wild counterparts, it is dependent on humans. Therefore, captive animals with an adequate food supply are expected to have less nutritional stress. Although the total amount of food fed was the same, decreased fGCM concentrations were observed with increased frequency of feeding. The red panda has a relatively inefficient digestive ability to process bamboo, their principal food component, being able to extract only one quarter of the energy, and necessitating that they spend a large amount of time foraging [32]. Thus, frequent feeding appears to be an essential parameter to reduce frustration and stress in captive red pandas, as reported in other species [10]. For example, incorporating frequent feeding regimens was better for the welfare of zoo elephants [33] and reduced stereotypic behaviours in captive Asian elephants [34]. In free ranging African elephants, the relationship between rainfall, a proxy for food availability, and fGCM concentrations was negative [35]. Similarly, a study on chimpanzees showed a negative association between monthly fruit abundance (food availability) and urinary cortisol concentration [36]. Our results further strengthen earlier findings that a more constant food supply through frequent feedings can potentially support good welfare in captive animals.

The present study showed that enclosure size and number of nests were associated positively with fGCM concentrations, which was unexpected, but could be confounded by the lack of enrichment in those spaces. Enclosure size for any species in a zoo setting should take into consideration the home-range size in the wild [37], which for red pandas is estimated to be 9.6 km^2^ [38,39]. Keeping enclosure size adequate to allow animals to roam and engage in species-specific behavior is important. However, not all studies report an effect of enclosure size on behaviour or cortisol/stress levels of captive animals [10]; i.e., in great apes [40]. In a study of caged long-tailed macaques, enclosure size had a minimal effect on urinary cortisol [41]. Similarly, housing rhesus monkeys in larger cages with more space did not reduce stereotypic behaviors [42]. In one study, transferring a young gorilla to a larger and more naturalistic enclosure actually resulted in an increase in some stereotypic/stress behaviours [43]. In another study, when margay individuals were treated with different enclosure conditions—large enriched, small barren and small enriched—the corticoid concentration increased respectively [44], suggesting that enriched but small enclosures are of no use in controlling physiological stress. Similarly, hiding places such as nests are important characteristics of the enclosure for captive animals to cope with stressors [10]. For instance, small felids used hiding places in response to elevated physiological stress [45] and stress behaviours reported in other studies [46,47,48,49]. In our study, the larger enclosures were deprived of trees and proper enrichment, and the enclosures that had more nests were lacking in size. Thus, providing space alone may not be sufficient to reduce or alter adrenal steroid activity in red pandas. While it is possible that red pandas in the larger enclosures were more active, something that can be associated with increased adrenal activity, there was no evidence of that in this study based on daily behavior observations. It is more likely that the sparse enrichment features in the small enclosure or vice versa inhibited natural behaviors during the day.

Under free-ranging conditions, the environment or habitat conditions can alter physiological fitness, while in captivity (i.e., zoos), enclosure characteristics and enrichment can play an important role in influencing physiology [50]. In the wild, the red panda is an arboreal species and well known to use trees and dead wooden logs for day-to-day activities such as feeding, resting, and moving, including constructing nests for parturition and rearing offspring. The presence of trees, dead wooden logs, and nests could therefore be an important cue for normal physiological responses [51]. Red pandas use fallen logs and shrubs to elevate themselves for several activities in the wild [52], making them crucial enrichment items in captivity. Consistent with these conditions, we found that providing more logs in the enclosure supports welfare in captive pandas in that they showed lower fGCM concentrations. These results are consistent with our previous findings that under captive conditions, enclosures that contained larger numbers of trees promoted better physiological function among red pandas, hence space or enclosure enrichment is an integral part of red panda welfare management [19].

We found that the fPM concentration was negatively affected by the enclosure area, which was not a surprise, as the enclosure area had a positive relationship with the fGCM concentration, as we discussed earlier in this section, and thus it reflects correctly in the fPM. For instance, in tigrinas, when subjected to different enclosure types—big-enriched, small-barren, and small-enriched—the concentration of fecal estrogens was higher in big-enriched enclosures and lowest in small-enriched enclosures [44]. This indicates the importance of providing enough enclosure areas with adequate enrichments to support welfare and reproductive fitness in captive red pandas. 

Finally, we found that fPM and fGCM concentrations were inversely related when controlled for tree density. This result suggests a potential negative effect of elevated glucocorticoids on reproduction, as has been shown previously [53]. In another study in captive female African elephants, serum progesterone was found to be negatively associated with cortisol [54]. Breeding records (Appendix A) suggest the success rate in previous years was only 50%, with three adult, reproductive-aged pandas having never given birth. Thus, further studies on how stress may be impacting reproduction in relation to management in pandas in Indian zoos are warranted.

## 5. Conclusions

Our study measured fGCM, fPM, and fAM concentrations in captive red pandas and demonstrated the effect of environmental and biological factors on steroid concentrations in pandas housed in a zoo setting. The fGCM and fPM results indeed suggest that providing appropriate enrichment, such as nests and logs, increasing the number of feedings, providing adequate enclosure areas, and controlling visitor numbers, could reduce stress and support better reproductive fitness in this species. However, the last of these, controlling visitor numbers, will be the most difficult to accomplish. Thus, providing structures and nest boxes that allow animals to hide if under duress are husbandry practices that a zoo could control, and it is where a zoo’s efforts should be focused.

Although results are preliminary due to low sample numbers, our study is the first to demonstrate relationships between fGCM and fPM, indicating a potential negative influence of stress on reproduction in captive red pandas. Lastly, we understand that a continuous sampling is key to studying the hormonal cycle in animals. We acknowledge that two samples from each subject is a major limitation, hence a long-term study with continuous sampling is needed to further confirm these preliminary results.

## Figures and Tables

**Table 1 animals-13-01298-t001:** Details of biological and environmental covariates collected during the study.

Name of the Covariates	Description
Biological covariates	
Sex	Male, female
Age	Obtained from zoo records
Sociality	Managed alone (single) or in a pair during at least part of the study (paired)
Social time	Length of time paired individuals were housed together (months)
Environmental covariates	
Number of visitors	Number of visitors at each enclosure recorded during observations
Ambient temperature	Measured using a digital thermometer every half hour during behavioural observations and when fecal samples were collected
Enclosure area (m^2^)	Measured as the two-dimensional area of the enclosure
Mean tree height used (m)	Mean height of living trees available to the pandas measured using a digital range finder
Log density (m^2^)	Number of logs (deadwood) placed on the ground or made into climbing structures within the enclosure as enrichment, counted and divided by the total the area of each enclosure
Tree density (m^2^)	Number of mature live trees (≥20 cm girth at breast height) counted and divided by the total area of the enclosure
Frequency of feeding	Number of times animals were fed per day
Quantity of bamboo	Bamboo offered (kg)/day obtained from zoo records
Number of nests	Number of nests/dens within each enclosure

**Table 2 animals-13-01298-t002:** Generalized linear mixed model (GLMM) analysis of the effect of biological and environmental covariates on fecal glucocorticoid metabolites (fGCM) of captive red pandas at three Indian zoos. Significant differences are in bold.

Predictors	fGCM (ng/g)							
	*F*	*df*	*p*	*Condition*	*Estimates*	*S.E.*	*t*	*p*
Sex (f/m)	4.32	1	0.073	female	22.47	12.81	1.75	0.073
Age	0.49	-	0.500		0.49	0.85	0.57	0.500
Social time (<1 month/≥1 month)	8.47	1	**0.021**	>1 month	−26.57	10.50	−2.52	**0.021**
Ambient temperature	3.55	-	0.073		3.00	2.13	1.41	0.073
Visitor number	31.70	-	**<0.001**		0.45	0.10	4.70	**<0.001**
Mean tree height used	0.46	-	0.506		−1.22	1.25	−0.98	0.506
Number of nests	17.42	3	**0.005**	2 nests	21.41	15.89	1.35	0.232
				3 nests	58.07	16.05	3.62	**0.019**
				4 nests	111.42	49.10	2.27	0.066
				1 nest	0	0	0	
Enclosure area	11.28	-	**0.011**		0.02	0.009	2.56	**0.011**
Tree log density	9.45		**0.018**		−5.56	2.72	−2.04	**0.018**
Frequency of feed (once/twice)	11.31	1	**0.010**	twice	−86.45	35.43	−2.44	**0.010**
Bamboo (<3 kg/≥4 kg)	0.88	1	0.371	≥4 kg	9.93	17.24	0.58	0.371
Sex: Subject	1.30	-	0.289		−1.03	1.03	−1.00	0.289
Subject: Enc. ID	2.14		0.177		0.42	0.35	1.21	0.177
**Random Effects**								
σ2	128.64							
τ00 Subject	3.11							
τ00 Enc. ID	0.00							
N _Subject_	20							
N _Enc_. _ID_	11							
Observations	39							
Marginal R^2^/Conditional R^2^	0.735/0.741							

**Table 3 animals-13-01298-t003:** Generalized linear mixed model (GLMM) analysis of the effect of biological and environmental covariates on fecal progestagen metabolites (fPM) of captive red pandas at three Indian zoos. Significant differences are in bold.

Predictors	fPM (ng/g) “Enclosure Effect”							
	*F*	*df*	*p*	*Condition*	*Estimates*	*S.E.*	*t*	*p*
fGCM	0.07		0.807		2.14	8.38	0.26	0.807
Mean tree height used	0.20		0.894		10.76	77.58	0.14	0.894
Number of nests	2.88	2	0.140	2 nests	−268	2121	−0.13	0.904
				3 nests	−1319	1676	−0.79	0.467
				1 nest	0	0	0	0
Enclosure area	7.80		**0.032**		−1.14	0.40	−2.79	**0.032**
Tree density	2.67		0.153		−780	477.67	−1.63	0.153
Tree log density	0.82	2	0.400		−110	121.57	−0.90	0.400
Subject: Enc. ID	5.30		0.061		−104	45.46	−2.30	0.061
**Random Effects**								
σ2	286,403							
τ00 Subject	0.00							
τ00 EncID	0.00							
N _Subject_	9							
N _EncID_	9							
Observations	18							
Marginal R^2^/Conditional R^2^	0.472/0.472							
**Predictors**	**fPM (ng/g) “No Enclosure Effect”**							
	*F*	*df*	*p*	*Condition*	*Estimates*	*S.E.*	*T*	*p*
Age	3.07		0.114		150	85.24	1.75	0.114
Social time (<1 month/≥1 month)	0.70	1	0.425	≥1 month	−331	396	−0.83	0.425
Ambient temperature	0.22		0.654		35.29	76.10	0.46	0.654
Number of visitors	1.40		0.268		−6.67	5.65	−1.18	0.268
Frequency of feed (once/twice)	0.30	1	0.595	twice	−288	523	−0.55	0.595
Bamboo (<3 kg/≥4 kg)	0.78	1	0.392	≥4 kg	452	509	0.88	0.392
**Random Effects**								
σ2	316,002							
τ00 Subject	0.00							
N _Subject_	0							
Observations	18							
Marginal R^2^/Conditional R^2^	0.392/0392							

## Data Availability

All necessary data supporting reported results are included as part of the manuscript.

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
