# Peer review of "Measures of Adrenal and Gonadal Hormones in Relation to Biological and Management Factors among Captive Red Pandas in Indian Zoos"

_animals, 2023, doi:10.3390/ani13081298_

Round 1
Reviewer 1 Report (Previous Reviewer 1)
The statistics are improved, but we're afraid they still aren't clear or correct.
- Zoo visitor number: Fig 1 makes it looks as though you have one value per animal, but this simply can't be correct.... No-one is monitoring how many visitors each individual panda is getting, are they? The methods make it sound more like you obtained one value for zoo, in which case Fig 1's x axis value should be in 3 bins, one per zoo. Fig 1 is also incorrect in that it presents an invalid simple regression that doesn't statistically control for sources of non-independence (like enclosure), other confounds (like sex), and it still treats the 40 samples as statistically independent when THEY ARE NOT (they come from just 20 animals in 12 enclosures). Please clarify how visitor number was obtained, and how finescale it was (e.g. per enclosure?); and just scrap Fig 1 and the related regression: it is simply not valid (no matter how pretty-looking!).
- Mann-Whitneys: are you double-counting your animals here, treating the 40 samples as independent even though they only come from 20 subjects in 12 enclosures? It's not clear. If you're pseudo-replicating, again that needs to be fixed (eg by averaging the two values per subject).
- The model at lines 232-239 is totally ad hoc: why control for tree density and nothing else? This looks like you went on a "P fishing expedition" behind the scenes! (Not valid - see all the literature on the dangers of P hacking).
- The use of the GLMM is definitely an improvement. However, there are six outstanding issues here.
First, it's the RESIDUALS from the model that have to be normal, not the raw data. If you checked the normality of the RESIDUALS then maybe then the reproductive data would prove amenable to this type of stats approach.
Second, the residuals also need to show homogeneity of variance. Did you check for this? This needs to be reported (and corrected if there is a problem).
Third, when reporting the test statistics, you need to present the numerator and denominator df for every F ratio.
Fourth it's not clear what terms were in the model. Was sex? (It needs to be). Was enclosure? (It needs to be as animals sharing an environment are not independent). So individual was nested in sex and enclosure? Some would also argue that enclosure should be random, not a fixed, effect: please clarify what you did, and explain your decision here.
Fifth, you are over-fitting. For one thing, it's not right to just "chuck in" a load of predictors and see what happens, because the predictors themselves may well be inter-related. This simply isn't valid. (We know some zoo researchers do this, but zoo researchers often mis-analyse their data terribly). Furthermore, you only have 12 enclosures and yet you are fitting something like 11 terms. This will use up all your df. Together I think this explains why your enclosure size effect is so weird (and so at odds with what you find in the Mann-Whitney in Table 1) - we bet it's just an artefact of this weird model. Suggested solutions below.
Sixth, it makes no sense to present both Mann Whitneys and GLMMs for the same data.
Two possible solutions for the problem of the over-complex, over-fitted model. First, do what we suggested last time and first investigate how all your animal characteristics and covariates inter-related, before making strategic decisions about what is sensible to include in a model. Alternatively, try a systematic "model fitting" approach, such as a step-wise forward regression, the selecting as the final model(s) the one(s) with the lowest AIC value.
- Supplementary Table 2: Why present the intercept? Makes no sense. Are you meaning to present the coefficient of the slope? And any p values to add here?
- Supplementary Table 3: Why only 9 females when the N of subjects is 10?
We hope this makes sense. It's a rich dataset that you should be proud of. Now you're using GLMMs, hopefully the changes are now easy to make. Fingers crossed the results become clearer, too.
Author Response
Please see the attachment

Reviewer 2 Report (Previous Reviewer 2)
I congratulate the authors on a much improved manuscript – the additional analyses and text add greatly to the scientific soundness of the work.
The manuscript is well presented and very clearly written. There are however some changes that are required :
Line 42: ‘Model’ should be ‘modelling’
Line 76: ‘maybe’ should be ‘may’
Line 84: remove ‘the’ …..influence reproduction negatively….
Line 84 – 92: clarification is needed – were the same animals that participated in the current hormone study also observed in the previous study that identified stereotypical behavioural expression? If they were the same animals this would reinforce the idea that the raised cortisol is an indicator of stress. If they are not the same animals then, unfortunately, my original comment regarding the misinterpretation of raised cortisol as, by default, stress, still stands and changes to the manuscript as discussed in my first review comments need to be evidenced. See comments below.
Line 134: remove ‘in’……faecal identification……
Line 190: two p values are reported
Table 3 – not sure what the lines are for and the table runs of the page. Only thing in bold is random effects – is that correct? I think your P values are off the page.
Line 232-234. P0.895 is not a trend, it shows no relationship. Please review working here.
Line 234: should read ‘when controlling for tree density….’
Line 250: should read ‘housed’ not house
Line 257: reference to welfare here should be reconsidered. I think ‘may be associated with reduced fGCM’ is a more appropriate interpretation. One isolated measure cannot tell you much about welfare. Could the authors instead state ‘There may be ramifications regarding animal welfare here, and this should be explored further in future research’?
Line 272: remove current discussion on stress. The increase in fGCM when visitor number is high could be stress, it could be excitement, it could be that the pandas have just moved towards or away from the visitors. From the data collected and the results presented ‘stress’ is possible, but not conclusive. Simply stating this ‘may indicate a stress response’ will be sufficient.
Line 312: There should be discussion here on activity. Cortisol increases with exercise to increase metabolic rate. Therefore large enclosures may encourage more activity and locomotion which may account for the observed pattern of fCGM release.
Line 347-349: stress discussion is overly strong and there needs to be some caution applied here.
In the Discussion there needs to be discussion on the limitations of this study. The major limitation is that each animal was sampled twice – hormones have cyclical rhythms of release and therefore multiple samples from an individual over multiple days would be typical. The possible ramifications of this on your results and interpretation also need to be considered.
Author Response
Please see the attachment.

Reviewer 3 Report (Previous Reviewer 3)
Overall the manuscript is much improved. Two additional comments:
- Final paragraph of the introduction references earlier work by this group. Since this seems to be foundational research supporting this investigation a more detailed report of the findings of that study and how those data lead to this investigation would be valuable. I think if the authors could add a few more sentences summarizing specific findings it would greatly improve the introduction.
- Table 3 and 4: Missing variables from statistical output should be included in the table. I use a paper by Crockford et al. (2013) as a standard example of a great GLMM table. It is open access and can be viewed here:
https://royalsocietypublishing.org/doi/full/10.1098/rspb.2012.2765
Author Response
Please see the attachment.

Reviewer 4 Report (Previous Reviewer 4)
While some minor changes and improvements have been made, serious flaws remain. As noted in my previous evaluation, drawing conclusions on fecal progestagens based on 2 samples spanning several months is largely meaningless. Secondly, based on the supplementary material, it appears that 17 of 20 animals are at one institution. The addition of the 2 other zoos (one with a single, solitary animal in a large enclosure, and one with a pair in a small enclosure) does not contribute to the study and in fact may obscure meaningful findings.
Figure 1: It is unclear how # of visitors was assessed. If you categorized as low (<80) or high (>80), how are you able to have an x axis that uses interval numbers? Was this the number of visitors in a day? At the exhibit? At the zoo?
Line 84: delete "the" between "influence" and "reproduction"
Line 87: delete "the" between "explored" and "factors" and between "affecting" and "psychological stress".
Line 89: delete the comma after investigates
Line 119: register should be registrar (I assume)
Line 255: this contradicts your results and the newly added paragraph beginning on line 298 (which is correct). As written, you say that housing red pandas in larger enclosures has a positive effect on welfare but your results showed the opposite (higher fGCM in larger exhibits).
Line 268: It is not invariably and always the case that visitors have negative effects on animal welfare. Visitors can also be enriching. See for example Hosey 2008 Applied Animal Behaviour Science for a pivotal paper. Many more recent papers explore the visitor-animal relationships as well.
Line 327: change results to result and suggest to suggests
Line 344: delete comma after although
Round 2
Reviewer 1 Report (Previous Reviewer 1)
I feel sorry for the authors as you really are having to meet a lot of conflicting suggestions (and the journal is not helping because their editorial oversight is so weak). To us the paper is weakened by the removal of the reproductive hormones, because now it’s just a paper on faecal corticosteroids, which really are meaningless as welfare indicators. That’s because baseline cort can increase OR instead decrease or even stay the same in subjects who are chronically psychologically stressed (see refs below for example; hypocortisolaemia is also common in atypical depression). Baseline core levels thus simply are not a valid way to infer welfare: increased values may be ‘good’ rather than ‘bad’ in terms of welfare, so assuming that ‘high cort indicates psychological stress’ is just naïve.
Also, the stats still are not right.
1) The GLMM is the right approach because it controls for subject, enclosure and sex effects (and other sources of non-independence): this therefore recognises that datapoints must be statistically independent for the valid use of stats/the value calculations of P values (see any stats textbook for details). The reason the linear regression shown in Fig 1 is therefore NOT CORRECT is because it does NOT take these sources of non-independence into account: it is treating each of the 40 values, even the two that come from each animal, as statistically independent: CLEARLY INCORRECT AND A CLASSIC EXAMPLE OF PSEUDOREPLICATION. Figure 1 (and the associated stats) therefore has to go. IT HAS TO GO!!!
Not only is it wrong, but it’s not even consistent with the rest of the paper: if a GLMM is the right statistical approach (which is is), then you need to use this FOR EVERYTHING. You can’t just add worse, extra, incorrect stats methods to the correct ones because you like the resulting graph : )
2) Sex*subject means “sex interacting with subject’ in most notations, and you cannot possibly mean that, as that term would not run (because each subject is only one sex: it is not cross-factored with sex). I think you mean “subject nested in sex”, with ‘sex’ also being a term in the model. Yes? The latter gives you the effect of sex. Subject should also be bested in enclosure; please can you confirm?
3) The check for collinearity is a great addition, but if two terms are collinear you DO NOT kick out both of them! You just kick out ONE. (And this then solves the collinearity problem). So the model needs to be re-run with tree OR log density added back in.
Miller, G. E., Chen, E., & Zhou, E. S. (2007). If it goes up, must it come down? Chronic stress and the hypothalamic-pituitary-adrenocortical axis in humans. Psychological bulletin, 133(1), 25.
O’Connor, Daryl B., Eamonn Ferguson, Jessica A. Green, Ronan E. O’Carroll, and Rory C. O’Connor. "Cortisol levels and suicidal behavior: A meta-analysis." Psychoneuroendocrinology 63 (2016): 370-379.
Author Response
All the comments and suggestions of the Reviewer 1 are incorporated into the revised version of the manuscript.

Reviewer 4 Report (Previous Reviewer 4)
The authors have done a thorough job at attending to reviewers' comments and significantly revising the manuscript. I offer some minor editorial changes below:
Line 39: co-founder should be confound
Ine 64: add “and” between “age,” and “sex”
Line 71: I would add “concentrations” after “stress hormone”
Line 96: while you’ve added exhibit sizes for SHZP and GBPHAZ, you did not do so for PNHZP. Since it is a large enclosure, the size may be in hectares perhaps, but for consistency, it is important to include this.
Line 102: some wording problem here; change to “…one tree and a nest; the rest of the enclosure…”
Line 158: add “an” between “as” and “interaction”
Line 160, add a comma after subjects.
Line 162: I think it would read better to say: “with variance inflation factor (VIF)>10 and thus excluded this from the…”
Line 164: change “run” to “ran”
Line 169: add fGCM before “concentrations” for clarity
I suggest one addition (in the discussion and/or conclusion): the results indeed suggest that providing appropriate enrichment, increasing number of feedings, and decreasing visitor number, could reduce stress. However the last of these – decreasing or controlling visitor number—would be most difficult to accomplish. The other two (increased enrichment, climbing structures, nest boxes, and increased number of feedings) are husbandry practices that a zoo could control and so this is where a zoo’s efforts should be focused.
Author Response
All the comments and suggestions made by the Reviewer 4 are incorporated into the revised version of the manuscript and the response to reviewer comments and suggestions file is also uploaded below.

Round 3
Reviewer 1 Report (Previous Reviewer 1)
One last comment: I'm so sorry, but I am really saying the same thing over and over.
The correct model is the GLMM. You've used this very nicely. And because the correct model is the GLMM you CANNOT say things like this :As fPM and fAM were not significant in the GLMM, data were subjected to univariate analysis; Mann Whitney U test and Analysis of Variance. GLMMs are correct because they correct for / control for enclosure effects that are shared by more than one panda. Mann-Whitney and simple Anovas are simply wrong because they don't do this.
I do recognize that with the addition of multiple covariates in the GLMM, you are losing many degrees of freedom and so you are losing power. But there might be other solutions to this problem, such as keeping the basic model structure (which is correct) but exploring different groups of covariates separately (such as perhaps 'within enclosure' effects, separated from 'outside enclosure' effects like visitor number, since what happens within and outside the enclosure are unrelated and independent). Each of those models would then have a bit more power.
Author Response
All comments and suggestion of the Reviewer 1 (Round 3) are agreed and the present version of manuscript is revised accordingly.
Please see the attached Response to Reviewer file attached for more details.

This manuscript is a resubmission of an earlier submission. The following is a list of the peer review reports and author responses from that submission.
Round 1
Reviewer 1 Report
It was exciting to see this paper from at least two researchers whose work I’ve long admired (Drs. Brown & Nagarajan), and on such an important species too. The thoroughness of the data collection, and the good sample size of animals spanning 11 separate enclosures, is also excellent.
That said, I and the PhD student I refereed this had some concerns, primarily about the statistics, but also about some discrepancies in reporting across the manuscript. However, hopefully these are easily fixed, as the analyses are easy to run in the package you’re familiar with (and the results will then be far more robust).
Statistics:
- Proper units of replication
Because you are mostly analysing the effects of enclosure characteristics (albeit some zoo level ones too), and have repeated samples from each animal, “animal ID” needs to be set as a random effect, and nested within “enclosure”. You only have three zoos, and Zoo 2 only has one enclosure, so we advise just kicking ‘zoo’ out of all your models. Each animal is sampled twice, but obviously those do not represent independent datapoints, so you either need to average the two values per animal, or have them as “repeated measures” (again with animal ID as a random effect). Running a regression across all faecal samples as shown in Fig. 1 is just not correct, because it’s double counting each animal (it’s thus “pseudoreplicating”, cf e.g. https://esajournals.onlinelibrary.wiley.com/doi/abs/10.2307/1942661 and https://bmcneurosci.biomedcentral.com/articles/10.1186/1471-2202-11-5).
- Analysing your data twice
It’s not clear why the three hormones were analysed in a series of categorical tests (Table 1) and also in a series of regressions. This seems to be analysing the same data twice, which risks Type I errors.
Also many of the Ns for some of the categories are so tiny as to give you no statistical power (for example, just one animal in the ‘solitary’ group for fAM, and so on; it obviously doesn’t make any sense to have an N of 1 in any group in an analysis).
It’s also not clear how the various variables inter-relate: are any confounded? We suggest a pre-screen of all input variable to see how they inter-related, and an exclusion of any categorical factors in which any group size is below 4 (since they have no real power).
Reporting issues in the manuscript:
L38-40: fGCM were said to be measured in 12 F and 8 M red pandas, whereas fPM and fAM seem actually to be measured in 10 F and 7 M, respectively (in Table 2). Likewise at Line 207, here the sample size for adult females seems to not be correct (n=12). Is it females including cubs or only adult females? If adult only, correct to n=10. And at line 212, here the sample size seems to not be correct (n=8). Is it males including cubs or adult males? If adult only, correct to n=7?
Samples were collected Dec – Feb, and yet breeding occurred Jan – March. So were samples all collected outside the breeding season, or did some overlap with the breeding season, and was that taken into account? Not clear.
Table 1:
Age: Should be classified as cub, adult and old?
Sociality: Does this include females housed with cubs?
Enclosure area: define “small” and “large”
Tree height: define “low” and “high”(etc etc throughout the table).
At Lines 243-244, is it said that the reproductive rate of all individuals was less than 50%. But the Suppl. Table 2 shows 100% and 75% breeding success in 2 pandas. Make it clearer that you mean the AVERAGE rate is below 50% (it seems to be around 34%). Also what does “rate” actually mean? “a/b” does not help because a and be are not defined!
Also, again in Suppl. Table 2, the FGCM and fPM concentration should be given similar units as in Table 2 in the manuscript.
Other minor issues:
L79: Correct “next boxes” to “nest boxes”
L89-91 : it is said that this study investigates how environmental and biological factors were related to stress and reproductive hormones and those in turn were related to reproduction.
But the relationship of hormones to reproduction is not discussed: it would be good to see this more clearly developed.
L 133-134: “Individual red pandas defecate at the same place in the enclosure, which aided individual identification”: say how exactly, and maybe give a reference?
L 273: Correct wild cohorts to wild counterparts?
- 29-295: Could sex differences in FGM just reflect sex differences in excretion route? (As in mice:https://www.sciencedirect.com/science/article/pii/S0016648002006202?casa_token=Y--41R7U_bQAAAAA:MO8-9wDBo6khsjdyeQctvTnbM_IRGLorGlgXlfcIPKbtk7HP-_a9bcvSPn9n5A1wOEP6A_IrgSU)
Reviewer 2 Report
This research area is much needed and given the species this is an original study.
The manuscript is very well written and presented.
There are several errors in methodological approach and interpretation of results which are discussed below.
Line 27-30 – can welfare be commented upon realistically? Are the animals with high cortisol excited or stressed? Please can the authors clarify for the reader.
Line 35: scientific name should be presented after common name the first time the common name is used.
In the abstract, can the assay type be briefly discussed as currently the methods of sample collection and assay are missing.
Line 54 –is there genuine evidence of stress – because there are no behavioural measures to verify that the animals appeared stressed rather than excited – both of which would be indicated by increased cortisol. Can the authors show that stress was evidenced during data collection specifically? If no behavioural date were collected the wording around stress needs to be softened – it may be stress rather than it was stress.
Line 65: ranges should be singular, range.
Line 82: does not read correctly – Climate factors may be a factor……..
Line 167-171 – can the comparisons that were analysed with Mann Whitney etc be listed to make the analysis more repeatable?
Was the data from one individual – 2 repeat samples, averaged before the rest of the analysis?
The analysis needs further consideration – the critical p value should be 0.001 or lower given the number of comparisons and tests performed.
There are issues with taking only 2 samples from each individual. The methods need to be extended to take multiple samples from all individuals during the same time period. The daily, yearly rhythms of cortisol and androgen release mean that two isolated samples from an individual tell one little about that individuals state of being. This needs to be acknowledged, discussed and is not. The methods should be extended and the analysis re-done with more reliable and valid individual measures of hormone.
Some comparisons involve highly unequal numbers of individuals – this is a massive issue in all analyses and limits the validity of comparisons. This is not discussed in the manuscript and needs to be.
The regression analyses explained low levels of variance – sometimes less than 50%, very low given the number of potential predictors included in the study. This is stated but not explored or discussed – there must be other variables influencing androgen production. Given the very low variance accounted for, the regression really suggests there is little to consider worthwhile in the sex hormone analyses particularly. This is not a reflection on the authors, it is just the way it is and this must be explored in the discussion.
In addition, reference to stress response is made throughout the manuscript and conclusions on welfare and husbandry made. High cortisol does not necessarily amount to evidence of a stress response – the animals could equally be excited, or more active or have just eaten etc. This reinforces the issues with having just two samples from each individual. There are no true baseline measures in the study and no behavioural data has been collected to verify if the individuals were stressed, more active, excited etc. This needs to be urgently addressed and reference to stress and welfare either removed or considerably softened – stress may be a factor here but there is no data to suggest it is a factor.
Reviewer 3 Report
This study compared fecal corticoid and fecal progestin/androgens in red pandas living in zoos. Overall, the approach and goals of this study have good purpose, however, I have concerns with the analysis and presentation of the data/study context.
Context/Presentation: The introduction does not provide sufficient context for the study. While I appreciate a concise and to the point introduction, the introduction ends with 18 citations, yet the paper has over 50. The majority of these citations are provided in the discussion, with little to no explanation of their purpose, study design or findings. This makes the points the authors are attempting to make very challenging to understand and appear vague/unclear. If these findings and points had been described in introduction, the ability to discuss their findings in the discussion section would be much improved.
Context/presentation: The storyline and discussions within this manuscript bounce between applied zoo animal welfare and field ecology. While I appreciate a broad discussion, it makes the paper feel vague and a little disorganized. I think there is value in interpreting the data from both points of view, but overall there is not a strong or consistent balance between the two. I think the points the authors are making in the discussion need to be prioritized better and have a clearer direction for interpretation.
Discussion: the points made within the discussion section are vague and more of a literature review than a true discussion (see point above additionally). For example, in several places paragraphs conclude with a general statement along the lines of “which agrees with earlier studies” or “has been shown previously.” This is not a strong or meaningful discussion of the study results and should be more specific.
Analysis: The analysis needs to be redone or explained in more detail.
- For example, the authors collected repeated measures data – 2 samples per red panda – but do not describe how they accounted for this. If they simply averaged the data then they have lost significant variability. Why was a generalized linear mixed model not utilized here that could handle the repeated measures and non-normality highlighted by the authors?
- The REVS test is not well known to me, however, doing some reading for this I feel like I have a better understanding. Though I still feel a GLMM would better fit the authors data. The authors need to provide citations and stronger justification for their use of statistics here, especially if the REVS test is less well understood by a general scientific audience. Additionally, the layout of your REVS table is unclear for your last three variables. How do they apply to each model (as they align differently in each section)?
- Independent variables: I feel like there is concern that some of the independent variables may be associated with one another. Was any testing done to assess for collinearity? For example, enclosure size, tree height, log density, tree density seem like they may all be high dependent on each other. Even if not statistically in terms of collinearity, it may be worth to better describe the variability between your enclosures and see where overlap is to best identify independent variables to include. To this point, only 2 pandas are in a closed environment, I feel like that may skew the distribution of some of your variables. Do you have sufficient sample size to analyze all of your variables?
- A heavy emphasis was put on visitor effect, which is an interesting topic. However, you only account for visitor numbers. There is no mention for proximity to pandas or design features that may affect this. For example, pandas in the closed exhibits may have a very different experience than the pandas in the 3000m2 exhibit if they can get quite a distance away. Numbers alone here may not be sufficient a variable. Is there additional considerations you can make here? Additionally, in your ethogram, you have two numbers of visitor level. How are these two related to your fecal samples? How are these numbers related to the overall guest experience for each panda group. More explanation and detail of how you assessed visitor presence and quantified it within your study are needed.
Reviewer 4 Report
This study investigates the impact of various environmental and biological factors on stress and reproductive hormones in red pandas housed in 3 zoos. Based on the supplementary material, it appears that all but 3 of 20 pandas are at one institution. This important fact was not mentioned, other than in the supplementary materials, but seems that it could be particularly important. While there was no significant effect of zoo, the fact that 17 animals were at zoo 1, 1 at zoo 2, and 2 at zoo 3, is clearly an imbalance, especially given the large individual variation that seems to be present. I realize this is the only way to address the environmental factors, but comparing 1 individual to 17 seems wildly inappropriate. Individual should probably be included as a random factor in analyses.
In addition to this problem, I have concerns over the fact that progesterone concentrations were based on 2 samples over a 2 week period. Given the cyclical pattern of progesterone, there is a chance that a female sampled at the nadir of her progesterone (ie, during the follicular phase of her cycle) could have a progesterone value orders of magnitude below that of a female sampled at the apex of her progesterone (ie, during the luteal phase). Thus, differences in progesterone concentrations are dubious at best without a denser sampling regimen.
Some specific comments:
Line 72: “suggests” should be “suggest” (the word “data” is plural)
Line 84: when referring to offspring survival being higher in younger pandas, I assume you are referring to the age of parents? Please clarify.
Lines 86 and 92: be consistent in spelling (progestagen versus progestogen)
Line 157: was the antibody for fPM also from Coralie Munro? If not, please indicate source.
Lines 173-175: There are many spelling errors in this sentence, making it difficult to comprehend. Please clarify.
Line 180: The assumption that stress is the independent variable causing changes in reproductive hormones may not necessarily be the case; the directionality could go the other way. For example, high levels of testosterone in males could lead to elevated cortisol as the breeding season progresses.
Lines 188-189: the difference in fGCM between males and females is indicated to be significant. Is there a reason this was not included in table 2? It is puzzling, given that the means (49.19 for males and 40.56 for females) represents a much smaller difference than, for example, age (42.31 for adults and 53.63 for cubs). I would like to see these results included in the table ad explained more fully. Also, the range in fGCM was enormous - 2.84 to 198.91!
Figure 1: does the significant finding still hold if the one outlier (visitor # 160, fGCM 200) is removed?
Line 292: promiscuous and polygynous are different mating systems; I assume you mean polygynous so please clarify.
I would suggest summarizing the findings with a few key recommendations for zoos. Given that changing exhibit size is often not possible, recommendations should include
- Increased number of feedings
- Increased number of nests
- Increased number of trees (where possible)